# Reversal of β cell de-differentiation by a small molecule inhibitor of the TGFβ pathway

Barak Blum[1]*, Adam N Roose[1], Ornella Barrandon[1], René Maehr[1†],
Anthony C Arvanites[1], Lance S Davidow[1], Jeffrey C Davis[1], Quinn P Peterson[1],
Lee L Rubin[1], Douglas A Melton[1,2]*

[1]Department of Stem Cell and Regenerative Biology, Harvard Stem Cell Institute,
Harvard University, Cambridge, United States; [2]Howard Hughes Medical Institute,
Harvard University, Cambridge, United States

**Abstract** Dysfunction or death of pancreatic β cells underlies both types of diabetes.
This functional decline begins with β cell stress and de-differentiation. Current drugs for type 2
diabetes (T2D) lower blood glucose levels but they do not directly alleviate β cell stress nor
prevent, let alone reverse, β cell de-differentiation. We show here that Urocortin 3 (Ucn3),
a marker for mature β cells, is down-regulated in the early stages of T2D in mice and when β cells
are stressed in vitro. Using an insulin expression-coupled lineage tracer, with Ucn3 as a reporter
for the mature β cell state, we screen for factors that reverse β cell de-differentiation. We find
that a small molecule inhibitor of TGFβ receptor I (Alk5) protects cells from the loss of key β cell
transcription factors and restores a mature β cell identity even after exposure to prolonged and
severe diabetes.

*For correspondence: blum@
fas.harvard.edu (BB); dmelton@
harvard.edu (DAM)

Present address: †Program in
Molecular Medicine, University of
Massachusetts Medical School,
Worcester, United States

Competing interests: The
authors declare that no
competing interests exist.

Reviewing editor: Hideyuki
Okano, Keio University School of
Medicine, Japan

## Introduction

Dysfunction or death of pancreatic β cells underlies all types of diabetes. In the case of Type 1 diabetes, it is unknown whether the initiating cause of β cell destruction is an immune attack or a β cell pathology that instigates autoimmunity. β cell failure in type 2 diabetes (T2D) is thought to begin as a compensatory response to peripheral insulin resistance and eventually results in the loss of a mature β cell phenotype, without necessarily leading to β cell death (*Weir and Bonner-Weir, 2004*; *Weir et al., 2013*). The loss of a mature β cell phenotype, sometimes called de-differentiation, can result from exposure to high levels of glucose, lipids, and inflammatory cytokines (*Accili et al., 2010*). De-differentiation of β cells in the context of diabetes has been shown in vivo with the genetic disruption of key transcription factors, including *FoxO1* (*Talchai et al., 2012*) and *NeuroD* (*Gu et al., 2010*), and is also seen in isolated islets cultured in vitro on an adherent substrate (*Gershengorn et al., 2004*; *Weinberg et al., 2007*; *Russ et al., 2008*; *Bar-Nur et al., 2011*; *Bar et al., 2012*; *Negi et al., 2012*). In both the *FoxO1* knockout mice and obese diabetic (Lepr[Db/Db]) mice, de-differentiating β cells gradually lose insulin expression and begin to express progenitor-cell markers including Ngn3 and Sox9 (*Talchai et al., 2012*). Oxidative stress, also associated with T2D, inactivates the β cell specific transcription factors *MafA*, *Nkx6.1*, and *Pdx1*, again leading to the loss of mature β cell identity (*Guo et al., 2013*). β cell de-differentiation may represent a reversal of the normal ontogeny of β cells, or follow a different pathway, but it is clear that de-differentiation depletes the pool of functionally mature β cells in T2D patients (*Weir and Bonner-Weir, 2004*; *Weir et al., 2013*). It is not known whether there are stages of de-differentiation at which the cells can recover or re-differentiate back into fully mature β cells. The commonly used T2D drugs act by suppressing glucose production

**eLife digest** Diabetes is a condition that develops when the body does not produce or use a hormone called insulin effectively. Insulin helps fat and muscle cells absorb glucose from the blood, and so diabetes can result in high levels of blood glucose, which can cause strokes, blindness, and heart disease.

In healthy individuals, beta cells in the pancreas (a large gland located behind the stomach) produce insulin. The beta cells develop from endocrine progenitor cells, which are unspecialised cells that can either duplicate themselves or 'differentiate' to form one of the specialised cell types found in the pancreas.

In diabetic patients, however, certain stresses (such as an immune attack in type-1 diabetics or insulin-resistance due to obesity, pregnancy, or ageing in type-2 diabetics) can cause mature beta cells to lose their identity in a process known as 'de-differentiation'. This means that beta cells either revert back to an earlier stage in their development or adopt a new dysfunctional identity. When this occurs, the body loses beta cells and is unable to produce insulin.

It was not known whether de-differentiated beta cells in diabetic patients can recover to form mature beta cells that are capable of producing insulin. Additionally, the drugs currently used to treat diabetes are able to lower blood glucose levels, but these drugs do not replace the lost beta cells.

Blum et al. now show that mice stop expressing a gene called *Urocortin 3* when beta cells first start to de-differentiate. Only functional beta cells express *Urocortin 3*, so this gene is a useful 'marker' that can be used to tell if a cell is a mature, functional beta cell or not. Using this system, Blum et al. found that if de-differentiated cells are transplanted into a non-diabetic mouse, they are able to revert back into mature beta cells that can produce insulin. This happens even if the cells have been de-differentiated for a long time.

Blum et al. then used this system to investigate ways of protecting against or reversing beta cell de-differentiation. Using small molecules to block the activity of a protein called TGF beta receptor 1 was found to protect against beta cell de-differentiation and to restore the identity of mature beta cells. The findings of Blum et al. represent a first step towards the possible development of new drugs to prevent or even restore the loss of healthy, mature beta cells in diabetic patients.

in the liver (e.g., Metformin), by enhancing peripheral insulin sensitivity (e.g., Rosiglitazone and other thiazolidinediones), or by forcing the secretion of more insulin from the already-stressed β cells (e.g., sulfonylureas such as Glyburide). There is no evidence that any of these drugs reverse β cell de-differentiation or restore the functionally mature β cell mass after β cell de-differentiation has occurred (*Kahn et al., 2006*; *Accili et al., 2010*). The availability of markers for early β cell stress (*Akirav et al., 2011*; *Mahdi et al., 2012*; *Erener et al., 2013*) allows one to test whether dysfunctional, stressed β cells can be revived or re-differentiated.

The gene *Urocortin 3* (*Ucn3*) is a marker for functionally mature β cells, cells capable of glucose stimulated insulin secretion (*Blum et al., 2012*). *Ucn3* expression appears relatively late in postnatal mouse development and its expression levels correlates with functional β cell maturation in mice, and with the maturation of human pluripotent stem cell-derived β cells after transplantation (*Blum et al., 2012*; *van der Meulen et al., 2012*; *Hua et al., 2013*; *van der Meulen and Huising, 2014*). We hypothesized that *Ucn3* expression may be lost or reduced early during β cell de-differentiation in T2D and if so, could be used to investigate the first steps of stress-induced β cell de-differentiation.

## Results

### Loss of Ucn3 expression is an early event in β cell de-differentiation in diabetes

Ucn3 and insulin expression in β cells of T2D mice were examined by immunostaining on pancreata of obese diabetic (Lep^{Ob/Ob} and Lepr^{Db/Db}) mice and from insulin-dependent diabetic mice (Ins2^{Akita}), and compared to pancreata of age matched non-diabetic (C57BL/6) mice. The intensity of insulin staining in diabetic mice is indistinguishable from non-diabetic controls, but the immunoreactivity of Ucn3 is almost completely abolished in islets of diabetic mice (*Figure 1A*). Quantitative real-time PCR

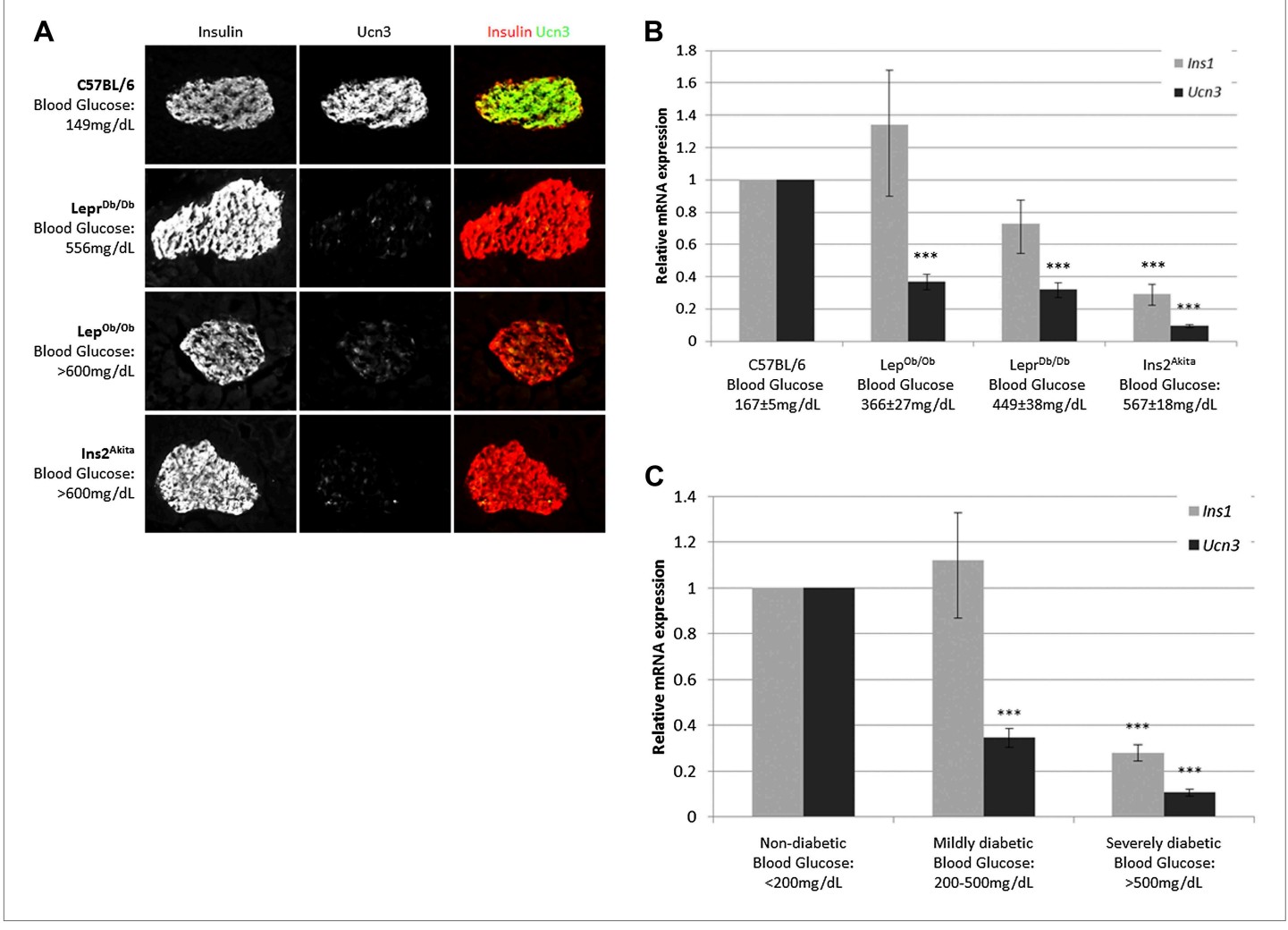

**Figure 1**. Loss of Ucn3 expression is an early marker for β cell de-differentiation in diabetes. (**A**) Immunostaining with antibodies against insulin (red) and Ucn3 (green) in pancreata from T2D (Lep[Ob/Ob] and Lepr[Db/Db]), insulin-dependent diabetic (Ins2[Akita]), and healthy control (C57BL/6) mice. Ucn3 protein but not insulin protein is down regulated in diabetic pancreata compared to the healthy control. (**B**) Quantitative Real-Time PCR analysis of *Ins1* and *Ucn3* gene expression in islets from C57BL/6 (*n* = 10), Lep[Ob/Ob] (*n* = 9), Lepr[Db/Db] (*n* = 8), and Ins2[Akita] (*n* = 11) mice. *Ucn3* mRNA is significantly reduced in all diabetes models, while insulin mRNA is significantly reduced only in the most diabetic model (Ins2[Akita]). (**C**) Quantitative Real-Time PCR analysis of *Ins1* and *Ucn3* gene expression in islets from non-diabetic control mice (*n* = 10; average blood glucose 167 ± 5 mg/dl), mildly diabetic (*n* = 16; average blood glucose 381 ± 17 mg/dl) and severely diabetic mice (*n* = 11; average blood glucose 588 ± 8 mg/dl). Error bars represent ±SEM. ***p < 0.001.

(qRT-PCR) showed that the expression of *Ucn3* mRNA levels is significantly (p > 0.001) reduced in islets of mice from all three diabetic models (*Figure 1B*). Statistically significant reduction in *Ins1* levels was only seen in the Ins2[Akita] mice, which also showed the highest fed blood glucose levels (*Figure 1B*). The disappearance of Ucn3 from β cells that still express high levels of insulin suggests that the loss Ucn3 is an early marker of β cell stress in diabetes, occurring before the reduction in insulin expression (*Talchai et al., 2012*; *Guo et al., 2013*).

Insulin expression has been previously reported to be diminished in β cells of severely diabetic mice, those with blood glucose levels exceeding 500 mg/dl (*Guo et al., 2013*). To confirm that loss of *Ucn3* is an early marker of diabetes, we divided the diabetic mice from all three models (Lep[Ob/Ob], Lepr[Db/Db], and Ins2[Akita]) into groups according to the severity of their diabetes, regardless of the genetic cause. Thus, the expression levels of Ins1 and Ucn3 mRNAs in the mildly diabetic (blood glucose levels between 200–500 mg/dl) and the severely diabetic (blood glucose levels >500 mg/dl) groups was compared to that of age-matched non-diabetic controls (C57BL/6, blood glucose levels <200 mg/dl).

The average (non-fasting) blood glucose level was 381 ± 18 mg/dl in mildly diabetic mice, 588 ± 8 mg/dl in the severely diabetic mice, and 167 ± 5 mg/dl in the non-diabetic control mice. The expression level of *Ins1* mRNA was slightly, but not significantly, higher in islets of mildly diabetic mice as compared to non-diabetic controls, but was reduced to 28% of control levels in islets of the severely diabetic group (p < 0.001). In contrast to the late reduction in insulin expression, the levels of *Ucn3* mRNA in the mildly diabetic group were already reduced threefold, to 34% of the level in the healthy control group (p < 0.001), and by 10-fold, to approximately 10% of the control levels, in the severely diabetic group (p < 0.001) (*Figure 1C*). We conclude that the loss of *Ucn3* mRNA is an early event in β cell de-differentiation.

## Using Ucn3 as a marker for the mature β cell state reveals reversibility of β cell de-differentiation

Because *Ucn3* expression is reduced early during β cell de-differentiation, its expression could be used to test whether β cells at early or late stages of de-differentiation are able to regain a fully mature state. The hypothesis is that while late-stage de-differentiated β cells (negative for both insulin and Ucn3) may not be able to re-differentiate into fully mature β cells, cells at an earlier stage (negative for Ucn3, but still expressing insulin) may be able to recover from their de-differentiation if the stress inducing factor (i.e., the diabetes) is removed.

To test this hypothesis, we induced transient insulin resistance in healthy, lean wild-type mice with the insulin-receptor antagonist S961 (*Vikram and Jena, 2010*; *Yi et al., 2013*). Mice treated with S961 develop acute insulin resistance and severe diabetes within 1 week, with non-fasting glucose levels of ≥500 mg/dl. Removal of S961 relieves the diabetes, and the mice restore their glucose control within 1 week. We thus induced transient hyperglycemia in wild-type mice with S961 for 1 week; control animals were similarly treated with PBS. At the end of the first week, half of the animals were sacrificed for analysis, and half were taken off S961 treatment and allowed to recover from diabetes for another week by which time their blood glucose levels returned to normal (≤200 mg/dl). Immunostaining of pancreata from all groups shows the levels of Ucn3 and insulin proteins (*Figure 2A*). As expected, animals treated with S961 developed diabetes (reaching blood glucose levels ≥460 mg/dl) and show an increase in insulin staining, while Ucn3 staining was almost completely abolished. In the diabetic animals that recovered and showed normoglycemia following withdrawal of S961 for 1 week, there was a complete recovery of Ucn3 staining, with a staining intensity comparable to that of the PBS-treated controls (*Figure 2A*). Quantitative RT-PCR analyses on islets at different time point during the development of S961-induced diabetes and its subsequent recovery showed that the levels of *Ucn3* mRNA are significantly (p > 0.005) reduced to about half of the levels in control mice as early as 4 days after S961 induction and are down to about a third by day 7 (*Figure 2B*). A small reduction of the *Ins1* mRNA was also seen, but this was not statistically significant. The expression of both *Ins1* and *Ucn3* increases to its normal levels (and even slightly higher) 3 days after pump removal, followed by a non-statistically significant decline 7 days after the withdrawal of the S961 pumps, which is not observed at the protein level (*Figure 2B,A*, respectively). A similar trend was seen with the expression of *MafA*, *Nkx6.1*, and *Pdx1*, confirming the loss and re-gain of the mature β cell state in this model (*Figure 2—figure supplement 1*).

We next tested whether more severely de-differentiated β cells can also return to a mature state after removal of the de-differentiation inducing stress. It has previously been reported that substantial β cell de-differentiation occurs when islets are cultured in vitro on an adherent substrate (*Gershengorn et al., 2004*; *Weinberg et al., 2007*; *Russ et al., 2008*; *Negi et al., 2012*). Cells de-differentiated using this method can be analyzed for the loss of their functional character and can be transplanted back into non-diabetic mice to test their differentiation state after being returned to a healthy environment (*Bar-Nur et al., 2011*; *Bar et al., 2012*).

In order to follow de-differentiated β cells, even after they cease to express insulin, we developed a lineage tracing system that marks cells that have expressed insulin (transcribed the insulin gene) in the past. *Insulin2-Cre* transgenic mice were crossed with mice carrying a floxed reporter of histone H2B fused to mCherry (*R26H2BCherry*), such that cells that had expressed insulin are marked with nuclear mCherry. These mice also contained a transgene driving cytoplasmic EGFP protein under the control of the *Ucn3* promoter (*Figure 3A*). The consequence of this genetic system is that cells with nuclear mCherry have, at some time, transcribed the insulin gene, but need not be actively producing insulin protein, and the (reversible) expression of cytoplasmic GFP indicates whether the β cell is fully

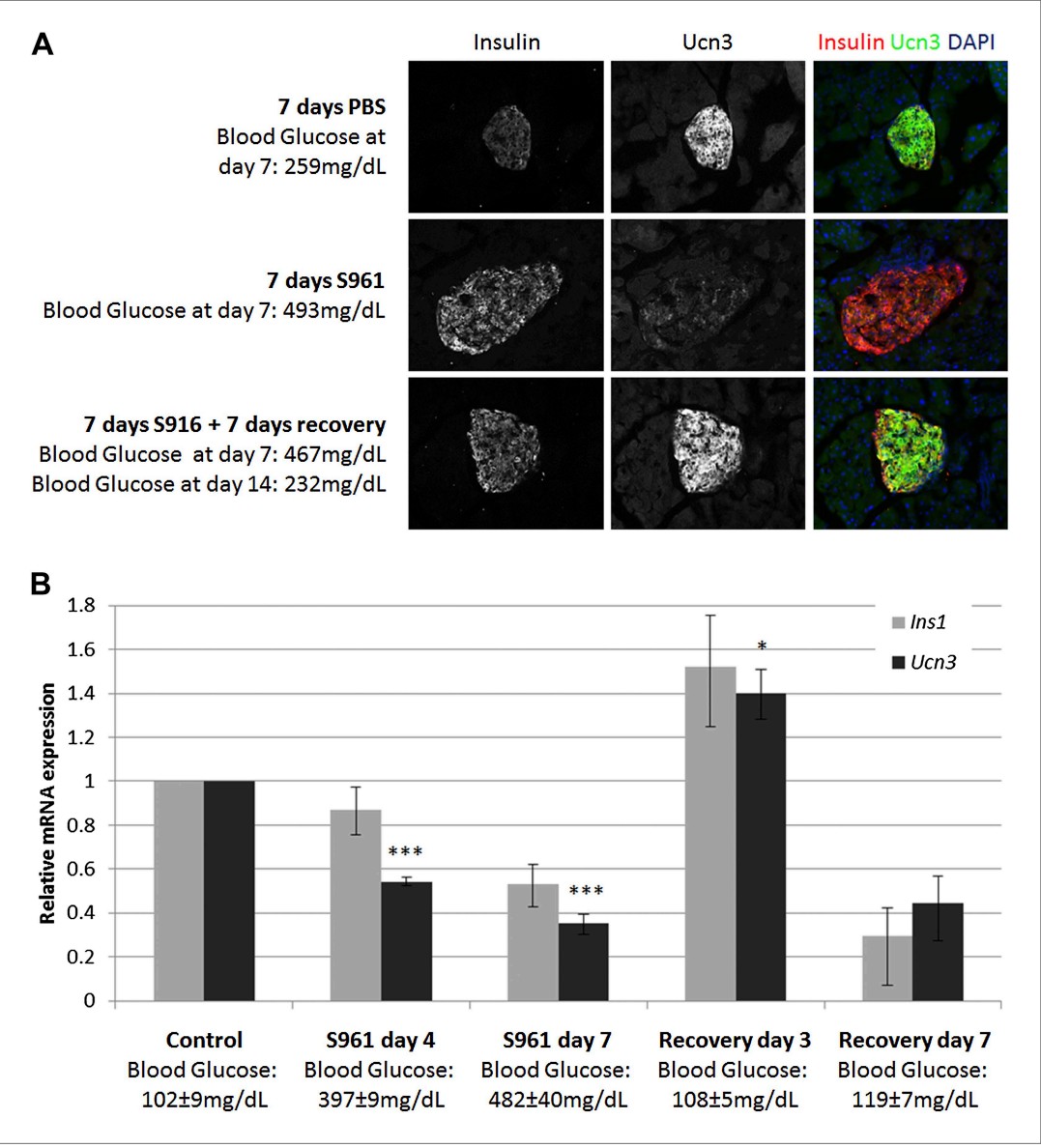

**Figure 2**. Insulin resistance-induced β cell de-differentiation is reversible. (**A**) Immunostaining with antibodies against insulin (red) and Ucn3 (green) in pancreata from wild-type C57BL/6 mice treated with either vehicle (PBS) or S961 (insulin receptor antagonist) for 7 days (upper and middle panels) or treated with S961 for 7 days followed by a 7-day-recovery period in the absence of S961 (lower panel). Ucn3 protein expression is down regulated in β cells following 7 days S961 treatment but returns to normal expression levels upon remission to normoglycaemia (see text). Nuclei are stained with DAPI (blue). (**B**) Quantitative Real-Time PCR analysis of *Ins1* and *Ucn3* gene expression in islets from ICR lean mice taken at different time points during S961-induced de-differentiation and post S961 withdrawal recovery (*n* = 3 mice for each stage). S961 osmotic pumps are transplanted on day 0 and removed on day 7. Control designates mice not treated with S961. Error bars represent ±SEM. *p < 0.05; ***p < 0.005.

The following figure supplement is available for figure 2:

**Figure supplement 1**. Expression of β cell genes during S961-induced de-differentiation and subsequent recovery.

mature (GFP positive) or de-differentiated (GFP negative). We labeled this genetic system 'RCU', for *R26H2BmCherry*; *Ins2*-Cre; *Ucn3*-GFP (***Figure 3A***).

Triple hemizygous RCU progeny are healthy and euglycemic (data not shown). The frequency of cytoplasmic *Ucn3*-derived GFP staining in all *Ins2*-Cre-derived H2BCherry-labeled cells was determined

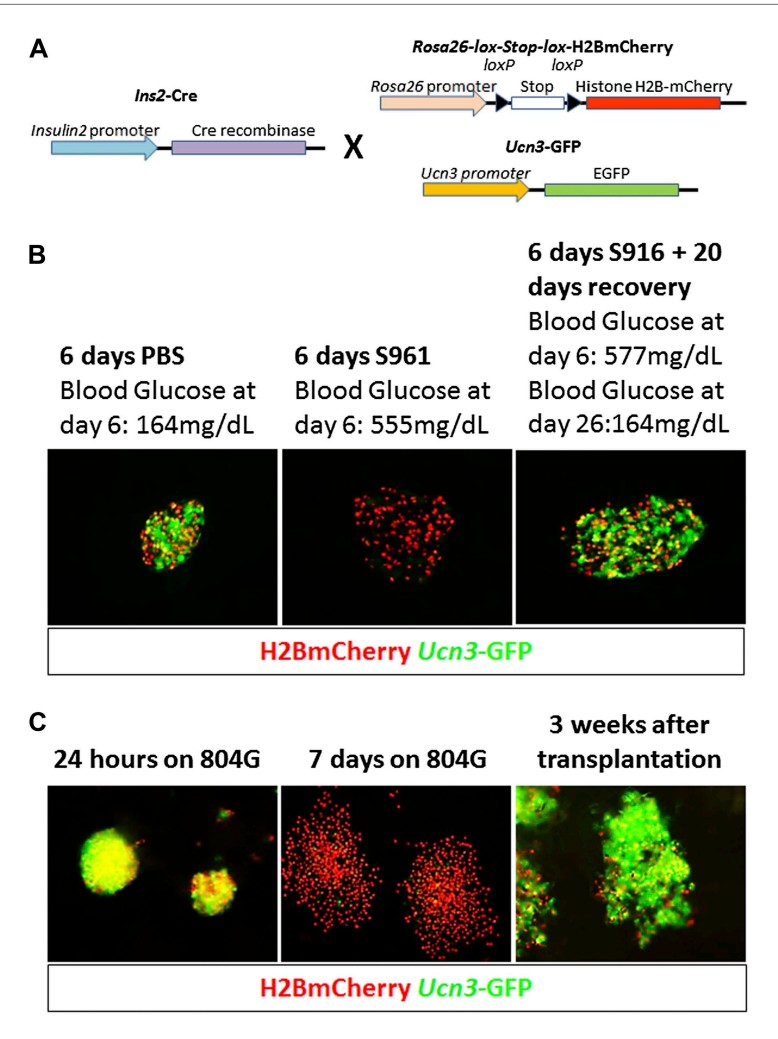

**Figure 3**. Adherent culture-induced β cell de-differentiation is reversible. (**A**) RCU reporter mice are made by crossing mice homozygous for the *Insulin2*-Cre transgene with mice doubly-homozygous for *Rosa26-lox-stop-lox-H2BmCherry* and *Ucn3*-GFP. *Insulin* expression in RCU progeny is permanently marked by red nuclear fluorescence, and *Ucn3* expression is marked by green cytoplasmic fluorescence. (**B**) Pancreas sections of PBS-treated control and S961-treated diabetic RCU mice. *Ucn3*-GFP is reduced in diabetic mice, but not in controls, and *Ucn3* expression returns after remission from diabetes. All images show live (unstained) reporter fluorescence. (**C**) De-differentiation and re-differentiation of RCU islets cultured in vitro. Islets from adult RCU mice were isolated and plated on 804G matrix for 1 week (left and middle panel). Note islet spreading and loss of *Ucn3*-GFP in the de-differentiated islets (middle panel). After 7 days, the de-differentiated islets were transplanted into euglycemic SCID mice for 3 weeks (right panel) after which time the transplants show the return of Ucn3 expression in β cells.

The following figure supplements are available for figure 3:

**Figure supplement 1**. RCU mice show nuclear insulin expression-coupled mCherry and Ucn3-derived cytoplasmic GFP.

**Figure supplement 2**. Ucn3 and insulin expression are down regulated in islets grown in adherent culture.

**Figure supplement 3**. β cells lose glucose-stimulated insulin secretion upon de-differentiation in culture.

by FACS to be 57 ± 16% in both male and female mice, between 1 month and 4 months of age (data not shown). Confocal imaging of β cells from triple hemizygous progeny RCU mice shows red nuclear fluorescence in β cells that is easily distinguished from the cytoplasmic green fluorescence emitted by the *Ucn3*-GFP reporter (*Figure 3–figure supplement 1*).

T2D-like symptoms were induced in RCU mice using the insulin antagonist S961 as described above. *Ucn3*-GFP levels are down-regulated in diabetic mice, treated with S961 for 6 days, but not in PBS-infused controls (*Figure 3B*, left and middle panels). After removal of S961, the expression level of *Ucn3*-GFP was up-regulated, returning to levels comparable to control animals (*Figure 3B*, right panel), corresponding to the remission of hyperglycemia (*Figure 3B*, right panel). These data show that loss of *Ucn3* expression is not permanent and that β cells can return to a mature Ucn3-positive state after a 7-day period of hyperglycemia.

When RCU islets are plated on an adherent 804G matrix (a laminin-rich extracellular matrix produced by the 804G rat epithelial cell line [*Lefebvre et al., 1998*]) and are cultured on the adherent matrix for 7 days, the islets flatten, cells spread out, and β cells lose *Ucn3*-GFP expression (*Figure 3C*, left and middle panels). The levels of both *Ins1* and *Ucn3* in such adherent cultures of islets from wild-type mice were reduced to 4% and 29% of the levels in freshly harvested islets, respectively (*Figure 3—figure supplement 2*). Consistent with the loss of the Ucn3 marker, these islets completely lose their ability for glucose-stimulated insulin secretion (GSIS, *Figure 3—figure supplement 3*). Most notably, the β cells re-express *Ucn3*-GFP 3 weeks after transplantation into the kidney capsule of euglycemic SCID mice (*Figure 3C*, right panel). These data suggest that the β cell de-differentiation caused by culturing cells *ex vivo* on adherent culture is reversible.

## Screen using de-differentiated RCU islets identifies roles for TGFβ pathway inhibitors and Artemin signaling in reversing β cell de-differentiation

The reversion of de-differentiated β cells to a mature state after transplantation to a healthy in vivo environment prompted us to look for factors that can recapitulate this phenomenon, as these factors could be candidates for drug development aimed at reversing β cell de-differentiation in T2D. We used the RCU platform to screen an array of 114 growth factors representing most major signaling pathways (*Supplementary file 1A*). The experimental design, outlined in *Figure 4A*, employs healthy islets from adult RCU mice, isolated on day 1 and plated on an adherent 804G matrix in a 384-well plate format. The islets were first cultured for 1 week to achieve adequate de-differentiation (see *Figure 3C* and *Figure 3—figure supplement 2*). Test compounds were then added on day 7 for another week. Each compound was tested in duplicate at two or three concentrations (listed in *Supplementary file 1A,B*). Fresh un-manipulated RCU islets were used as a positive control, and DMSO- or non-treated cultures were used as a negative control. The islets were fixed on day 11 for automated imaging and subsequent analysis. Percentages of mCherry positive cells that co-express GFP were calculated for each well and used to identify conditions that significantly increased the number of GFP positive cells over negative (DMSO- or non-treated) controls (*Figure 4A*). Positive hits were selected according to their statistical significance (p value) over the negative control. Of the 114 tested factors, three growth factors restored *Ucn3*-GFP expression with a high statistical significance (p < 0.01, *Figure 4B*). These factors are BMP9, soluble TGFβ receptor 3 (TGFβ sRIII, also known as betaglycan), and the GDNF-family member Artemin.

Both BMP9 and TGFβ sRIII signal through receptors of the TGFβ receptor family (*Massague and Chen, 2000*; *David et al., 2007*), whereas Artemin signals through RET and GFRα3 receptors (*Airaksinen and Saarma, 2002*). To delve deeper into the effects of BMP9, TGFβ sRIII, and Artemin on β cell re-differentiation, a second screen was performed using a library of 19 small molecule kinase inhibitors of TGFβ signaling and 18 small molecules effectors of RET/GFRα3 signaling (*Figure 4C* and *Supplementary file 1B*). In addition, we included a library of 42 known drugs for T2D (*Figure 4C* and *Supplementary file 1B*).

Among the 19 small molecules tested in the TGFβ receptor inhibitors group, Alk5 inhibitor I, Alk5 inhibitor II, and a SMAD3 inhibitor, restored *Ucn3*-GFP expression in de-differentiated β cells (p > 0.01; *Figure 5C*). Of the 18 RET/GFRα3 inhibitors two molecules with relatively low specificity to the RET kinase, namely PHA-739358 and VEGFR inhibitor V, induced Un3-GFP in the cells with p-values below 0.01, and of the 42 known T2D drugs, only the two potassium-channels blockers, Repaglinide and Tolbutamide, gave marginal results.

Alk5 inhibitor II showed the strongest effect among all molecules tested, both by its reproducibility (as measured by its statistical p value over DMSO-treated controls) and on the levels of *Ucn3*-GFP expression, restoring *Ucn3*-GFP fluorescence of de-differentiated RCU islets to levels comparable to that of fresh islets (*Figure 5A*). A dose–response test showed that its effect on *Ucn3*-GFP expression in de-differentiated RCU β cells begins at nanomolar concentrations (*Figure 5—figure supplement 1*).

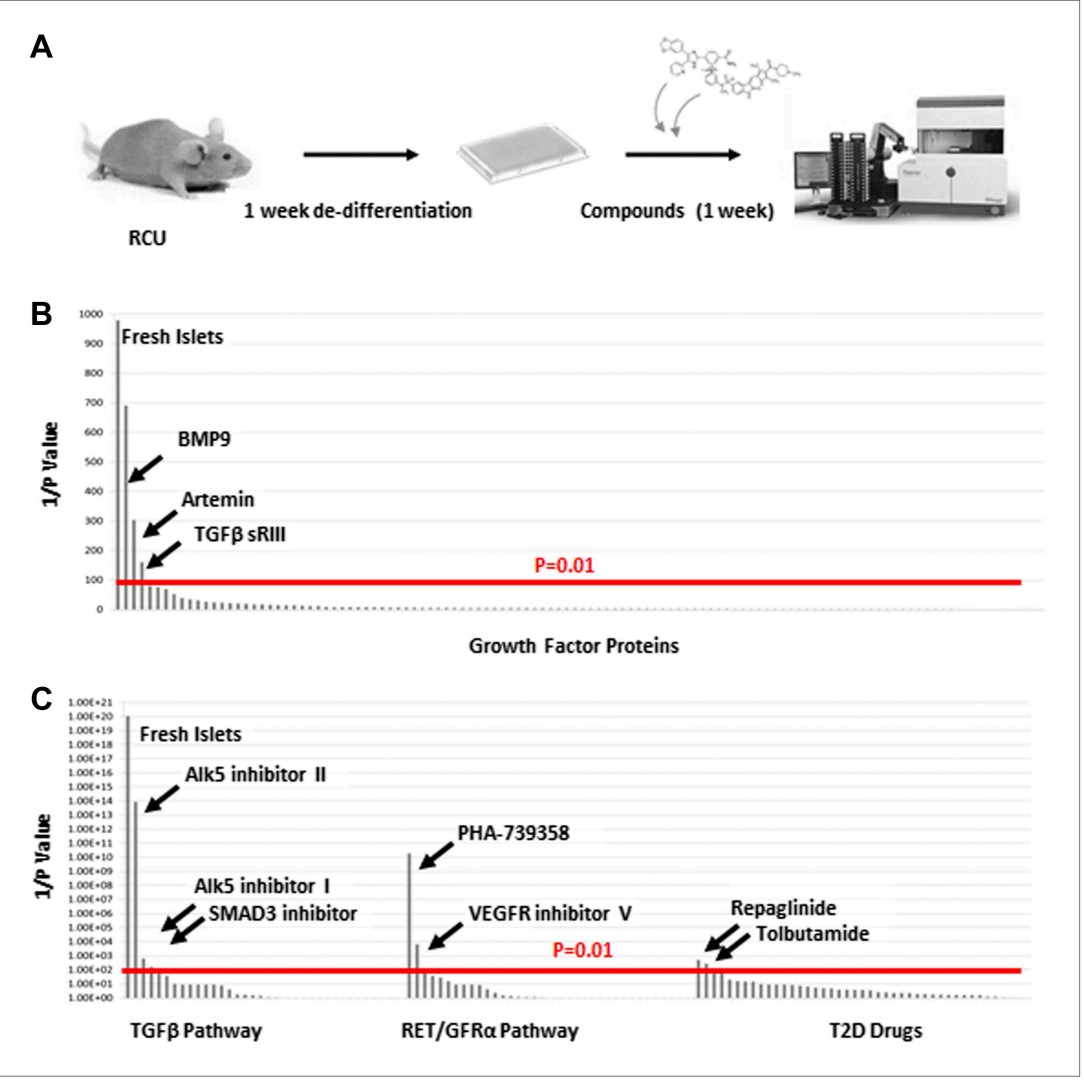

**Figure 4**. TGFβ pathway inhibitors and Artemin signaling reverse β cell de-differentiation. (**A**) Islets from adult RCU mice are isolated and plated on 804G matrix for 1 week in a 384-well plate format during which time the β cells de-differentiate. A compound library is added on day 7, and islets are cultured for an additional week in the presence of compounds. Each compound is tested in duplicates of two or three concentrations. Fresh un-manipulated RCU islets are used as a positive control, and DMSO- or untreated islets are used as negative controls. Islets are fixed on day 11 for automated imaging and subsequent analysis. Percentages of mCherry positive cells that co-express GFP are calculated for each well and used to identify conditions that significantly increase the number of GFP positive cells over negative (DMSO- or non-treated) controls. Positive hits are selected according to their statistical significance (p value) over the negative control. (**B**) Results of screen with 114 growth factor proteins. Factors are ordered from left to right based on the p-value of their *Ucn3*-GFP fluorescence over the negative (non-treated) control. For convenience, values on the Y axis are presented as 1/p-value. Red bar represents the threshold for statistical significance (p < 0.01). (**C**) Results of screen with 19 TGFβ pathway inhibitors, 18 RET/GFRα3 inhibitors, and 42 known T2D drugs. Factors are ordered from left to right based on the statistical p-value of their *Ucn3*-GFP fluorescence over the negative (DMSO-treated) control as above. For convenience, values on the Y axis are presented as 1/p-value. Red bar represents the threshold for statistical significance (p < 0.01). A full list of the factors tested is presented in the ***Supplementary file 1***.

To confirm the reviving effect of Alk5 inhibitor II, we performed qRT-PCR analyses on FACS-sorted mCherry-positive β cell from islets de-differentiated on 804G matrix for 1 week, followed by another week of culture on 804G matrix supplemented with Alk5 inhibitor II, and compared those cultures grown on 804G matrix without added Alk5 inhibitor II and those of freshly-isolated islets (***Figure 5B***). The

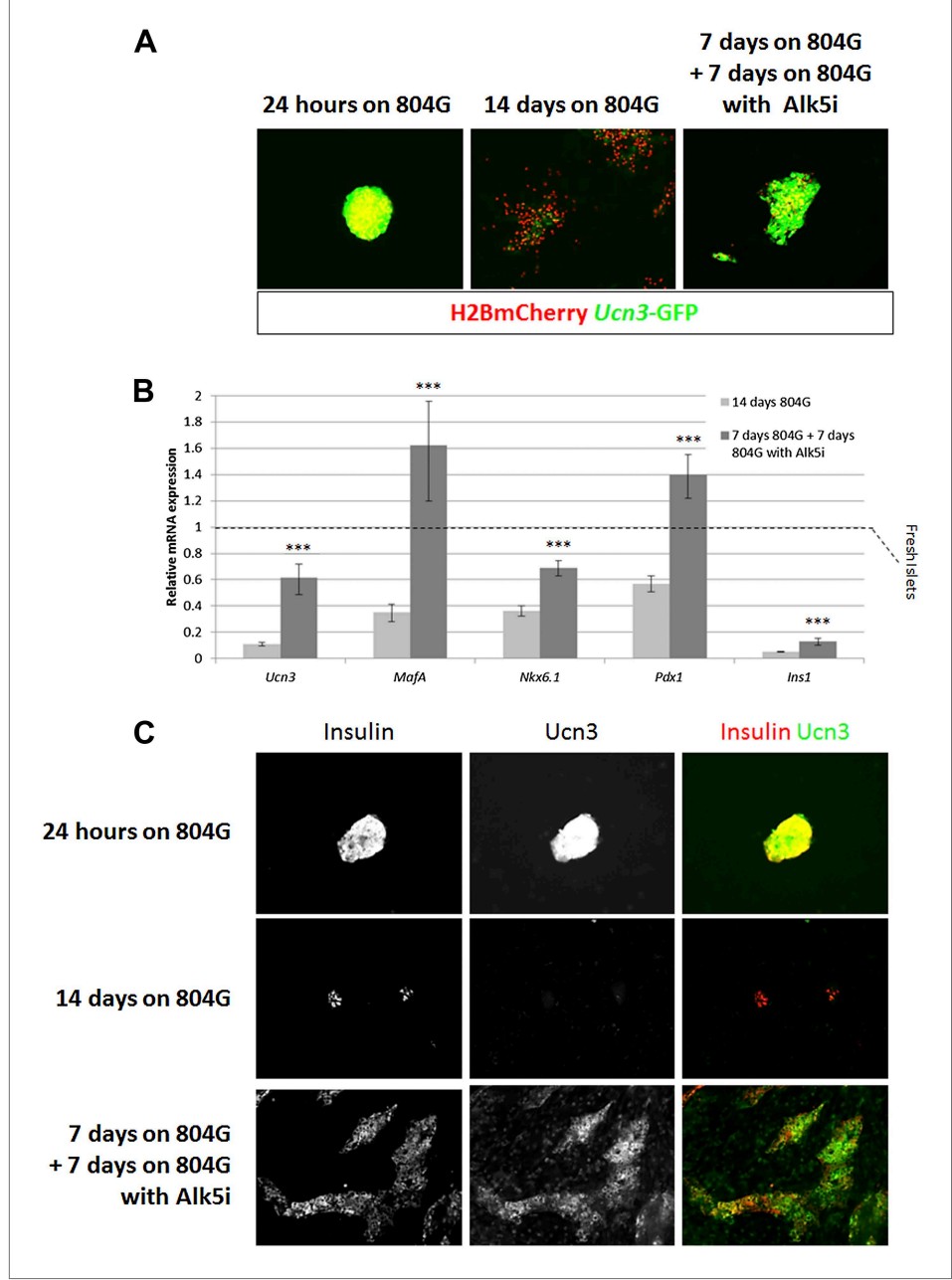

**Figure 5**. Alk5 inhibitor II restores β cell maturation in 804G-induced β cell de-differentiation. (**A**) Islets from adult RCU mice were isolated and plated on 804G matrix for 14 days with or without the addition of Alk5i at day 7 (right and middle panels, respectively). Live fluorescence images of H2BmCherry and *Ucn3*-GFP were taken on day 14, and compared to fresh RCU islets cultured on 804G for 24 hr. (**B**) H2BmCherry-positive cells from the above cultured were sorted by FACS and subjected to qRT-PCR analysis for the expression of various mature β cell genes. Statistical significance relates to the difference between Alk5i-treated and non-treated islets for each gene. Expression levels are normalized to the levels of freshly isolated islets (dashed line). Error bars represent ±SEM of three biological repeats. ***p < 0.001. B.G. (**C**) Immunostaining with antibodies against insulin (red) and Ucn3 (green) in islets from ICR mice treated as above.

The following figure supplement is available for figure 5:

**Figure supplement 1**. Alk5 inhibitor II induces *Ucn3*-GFP in RCU islets in a dose-dependent manner.

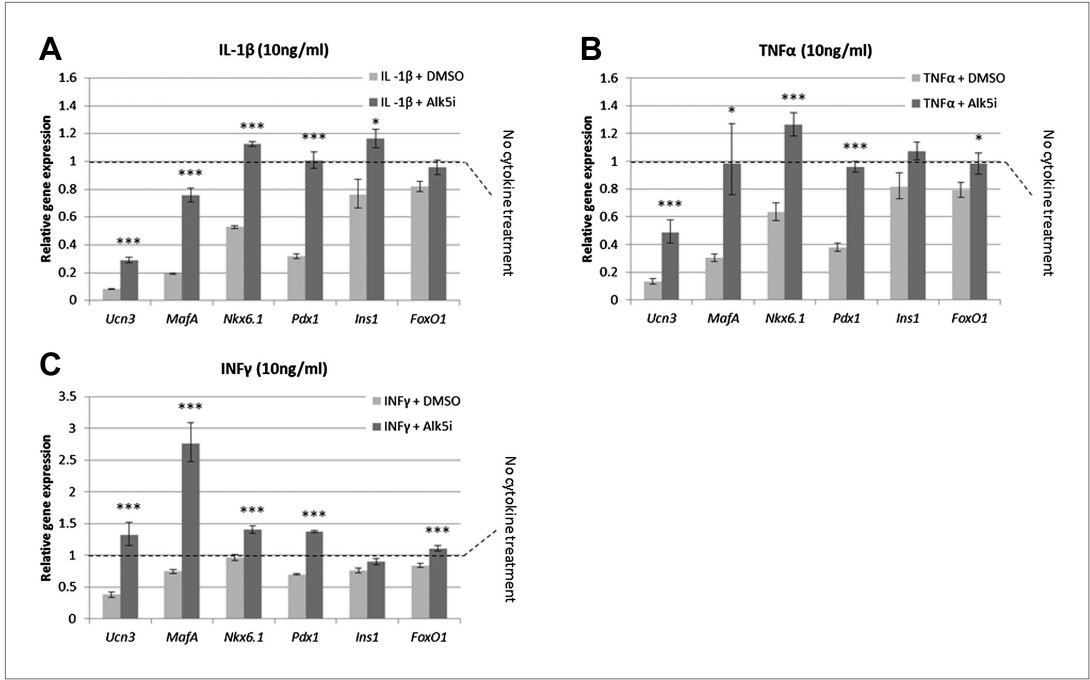

**Figure 6**. Alk5 inhibitor II induces expression of mature β cell transcription factors and prevents their reduction under cytokine stress. Quantitative Real-Time PCR analysis of gene expression in wild-type islets treated with cytokines as shown (**A**) IL-β, (**B**) TNFα, (**C**) INFγ. Each bar represents average gene expression in three independent experiments. Expression levels are normalized to the levels of control islets not treated with any cytokine (dashed line). Statistical significance relates to the difference between Alk5i-treated and DMSO-treated islets for each gene. Error bars represent ±SEM. *p < 0.05; ***p < 0.005.

The following figure supplement is available for figure 6:

**Figure supplement 1**. β cells lose glucose-stimulated insulin secretion upon cytokine treatment.

expression levels of *Ucn3*, *MafA*, *Nkx6.1*, *Pdx1*, and *Ins1* were severely reduced in cultures de-differentiated for 2 weeks on 804G matrix. Strikingly, addition of Alk5 inhibitor II for 1 week after the initial first week of de-differentiation significantly (p < 0.001) induced the expression levels of *Ucn3*, *MafA*, *Nkx6.1* and *Pdx1*, and in the case on *MafA* and *Pdx1*, to levels greater than those of freshly isolated islets (*Figure 5B*). The recovery of the expression of *Ins1* mRNA is also statistically significant, but its expression levels were still lower than those of fresh islets, perhaps because the experiment was done in low glucose medium. Immunostaining on islets from WT mice cultured as above confirmed the recovery of insulin and Ucn3 proteins in Alk5 inhibitor II re-differentiated β cells (*Figure 5C*). Nevertheless, the addition of Alk5 inhibitor II to 804G matrix-induced de-differentiated β cells was not sufficient to restore GSIS in vitro to a statistically significant level (data not shown), indicating that there is more to functionally mature GSIS than mature β cell gene expression.

## Alk5 inhibitor II up regulates expression of β cell transcription factors and prevents their loss under cytokine stress

It has recently been shown that under diabetes-related stress, the expression and activity of key β cell transcription factors, including *MafA*, *Nkx6.1*, and *Pdx1*, are compromised (*Guo et al., 2013*). We thus tested whether Alk5 inhibitor II is capable of preventing the down regulation in expression of these transcription factors. Islets harvested from lean, non-diabetic mice, were exposed to a diabetes-related cytokine challenge for 24 hr, with or without the presence of Alk5 inhibitor II, and the expression of several β cell genes was measured by qRT-PCR and compared to islets not treated with cytokines (*Figure 6*).

Islets exposed to 10 ng/ml of either IL-1β, TNFα or IFNγ showed abrogated GSIS response (*Figure 6—figure supplement 1*) and reduced expression of *Ucn3*, *MafA*, *Nkx6.1*, and *Pdx1* mRNAs,

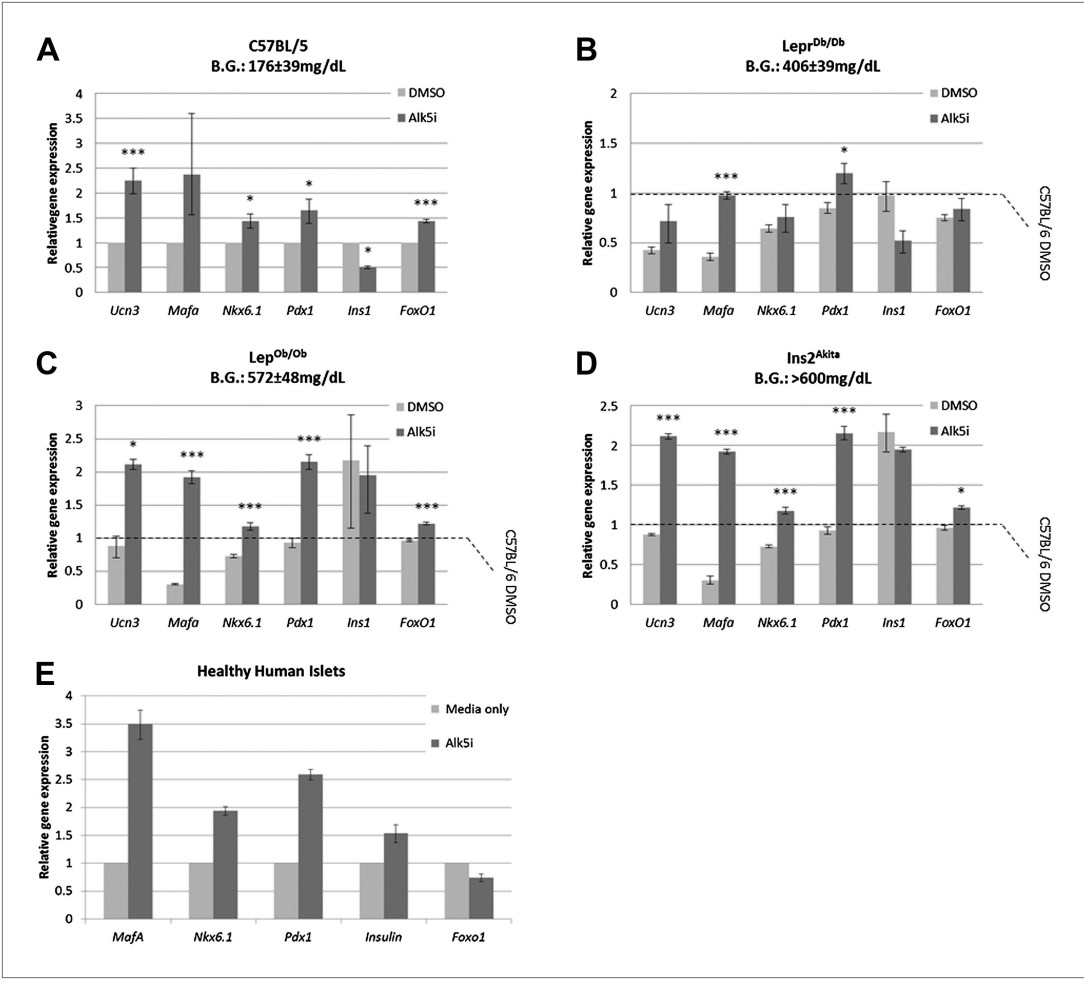

**Figure 7**. Alk5 inhibitor II induces expression of mature β cell transcription factors even in β cells that were exposed to extreme diabetic conditions for several months. (**A–D**) Alk5 inhibitor II (Alk5i) induces expression of specific β cell genes in islets from healthy and severely diabetic mice. Shown are quantitative Real-Time PCR analysis of gene expression in islets of healthy control (C57BL/6) and diabetic mice (Lepr$^{Db/Db}$, Lep$^{Ob/Ob}$, and Ins2$^{Akita}$). Each bar represents average gene expression in three independent experiments for each group. Statistical significance relates to the difference between Alk5i-treated and DMSO-treated islets for each gene. Expression levels are normalized to the levels of C57BL/6 islets treated with DMSO (dashed line). Error bars represent ±SEM. *p < 0.05; ***p < 0.005. B.G. = Blood glucose level at time of sacrifice. (**E**) Alk5 inhibitor II (Alk5i) induces expression of specific β cell transcription factors in human islets. Shown are quantitative Real-Time PCR analyses of gene expression. Error bars represent three technical repeats on islets from a single donor. Error bars represent ±SEM.

whereas expression of *Ins1* and *FoxO1* was less affected (*Figure 6*). Addition of 1 µM Alk5 inhibitor II with any of the cytokines prevented the diminution of expression levels for *Ucn3*, *MafA*, *Nkx6.1*, and *Pdx1*. In fact, the expression levels of the latter three genes remained at levels comparable to, and in some cases higher than, that found in control islets (those not exposed to cytokines) (*Figure 6*). However, as seen with 804G de-differentiation, the addition of Alk5 inhibitor II to cytokine-treated islets was not sufficient to restore fully functional GSIS (data not shown).

## Alk5 inhibitor II can restore expression of β cell transcription factors even in β cells that were exposed to extreme diabetic conditions for several months

We asked whether Alk5 inhibitor II can restore the expression levels of specific β cell genes from severely diabetic mice, β cells that were exposed to an extreme diabetic environment for several months. To answer this question, gene expression analyses were performed on islets from lean

non-diabetic C57BL/6 mice and from mice with advanced to severe diabetes (Lepr[Db/Db], Lep[Ob/Ob], and Ins2[Akita]; blood glucose levels of 406 ± 39 mg/dl, 527 ± 48 mg/dl, and >600 mg/dl, respectively). The islets isolated from these diabetic animals, and controls, were cultured in vitro for 24 hr with or without Alk5 inhibitor II (*Figure 7*). Culturing control healthy islets for 24 hr with Alk5 inhibitor II results in a 1.5- to2.5-fold higher expression of *Ucn3*, *MafA*, *Nkx6.1*, *Pdx1*, and *FoxO1* compared to DMSO-treated controls (*Figure 7A*). We also observe an unexplained twofold decrease in *Ins1* expression (*Figure 7*). Similarly, islets from mice with advanced diabetes (*Figure 7B*) and islets from mice with severe diabetes (*Figure 7C,D*) responded to the Alk5 inhibitor II. The increase in *Ucn3*, *Nkx6.1*, and *Pdx1* gene expression caused by Alk5 inhibitor II in severely diabetic mice was 1.5- to 2.5-fold, similar to the effect on non-diabetic islets. The induction of *MafA* expression by Alk5 inhibitor II in the severely diabetic islets increased to fivefold and sixfold over DMSO-treated controls (*Figure 7C,D*). This may reflect the early role of *MafA* disappearance in β cell stress (*Guo et al., 2013*). We conclude that Alk5 inhibitor II can induce mature gene expression in β cells that were exposed to extreme diabetic conditions for several months.

We also tested whether the Alk5 inhibitor II is effective in restoring specific β cell gene expression in human islets. Primary human islets were treated with Alk5 inhibitor II for 24 hr and subjected to gene transcript analyses (*Figure 7E*). Similar to the results with mouse islets, human islets treated with Alk5 inhibitor II show an increase in mRNA expression for Insulin, *MafA*, *Nkx6.1*, and *Pdx1* mRNAs, but not for *FoxO1* (*Figure 7E*).

## Discussion

The response of β cells to the progression of T2D begins with an adaptive stage, in which the cells compensate for insulin resistance by over-production and over-secretion of insulin, as well as increasing β cell replication (*Weir and Bonner-Weir, 2004*; *Guo et al., 2013*; *Yi et al., 2013*). This adaptation is reversible, as can be seen when β cell function returns with the remission from T2D after bariatric surgery (*Bradley et al., 2012*). However, if the metabolic stress persists, β cells succumb to the metabolic overload and de-differentiation occurs. De-differentiation begins with translocation of the transcription factor *FoxO1* to the nucleus, and continues with an inactivation of β cell-specific transcription factors including *MafA*, *Nkx6.1*, and *Pdx1* and consequently, a reduction in insulin production and secretion. All together, these changes result in the escalation of the disease and eventually to a non-recoverable loss of a functionally mature β cell mass (*Weir and Bonner-Weir, 2004*; *Talchai et al., 2012*; *Guo et al., 2013*).

Our results put the loss of Ucn3 expression as an early event in β cell stress, occurring at the compensatory stage, before reduction in insulin expression and deterioration to frank diabetes. Ucn3 protein and mRNA were dramatically down regulated even in mildly diabetic mice, some of which had blood glucose levels that were just slightly above normal. This is further demonstrated by the loss of Ucn3 in S961-treated mice, exposed to insulin resistance and hyperglycemia for only a week. In severely diabetic mice, we observed a dramatic reduction in insulin expression indicative of advanced β cell de-differentiation, and the expression of *Ucn3* mRNA was almost completely abolished.

Ucn3 is a small neuropeptide hormone, expressed mainly in the Islets of Langerhans (where, in the mouse, it is restricted to β cells), the small intestine, the skin, and specific brain regions such as the hypothalamus, amygdala, and brainstem (*Lewis et al., 2001*; *Li et al., 2003*; *Benner et al., 2014*; *van der Meulen and Huising, 2014*). Ucn3 has been suggested to regulate GSIS in response to high blood glucose (*Li et al., 2007*) and was linked to peripheral glucose homeostasis and food intake behavior (*Kuperman and Chen, 2008*; *Kuperman et al., 2010*; *Jamieson et al., 2011*). The precise role of Ucn3 in the pancreatic islets is not yet clear, but the expression of its receptors in mouse and human islets suggests an islet-autonomous autocrine and/or paracrine action (*Huising et al., 2011*). It is noteworthy, however, that loss of expression of Ucn3 per se is not a driver of diabetes, but is rather caused by it, as mice homozygous for a *Ucn3*-null allele are not diabetic, and even show slightly better glucose tolerance under high-fat feeding and aging (*Li et al., 2007*).

To utilize the finding that Ucn3 is an early marker of β cell de-differentiation, we developed triple-transgenic mice, in which a sensitive *Ucn3*-regulated GFP reporter is combined with β cell lineage tracing. This genetic system allows one to trace β cells even after profound de-differentiation. Using this system, we show that β cell de-differentiation can be reversed after 1 week of S961 treatment in vivo or after 1 week of adherent culture in vitro. A screen for pathways that can rescue β cells from de-differentiation identified three growth factors that restored *Ucn3*-GFP expression, namely BMP9,

TGFβ sRIII, and Artemin, all belonging to the TGFβ superfamily. Of those, only BMP9 had previously been identified as having an active role in glucose homeostasis (*Chen et al., 2003*). The gene encoding TGFβ receptor III has been shown to be up regulated in pancreata from obese human patients compared to lean subjects (*Muharram et al., 2005*), while Artemin and its receptor GFRα3 have, to the best of our knowledge, not been described in pancreatic islet function.

Tests on small molecule mediators of BMP/TGFβ and Artemin signaling identified Alk5 inhibitor II as a potent compound able to restore mature β cell identity even in islets from severely diabetic mice. This inhibitor also blocked the loss of specific β cell gene expression under cytokine-induced stress. Alk5 inhibitor II, identified using mouse β cells, can induce the expression of key β cell transcription factors in human islets. While human UCN3 is a marker of the functional maturation for both β and α cells (*van der Meulen et al., 2012*; *Benner et al., 2014*), and despite evidence that UCN3 is not a faithful marker for functional β cell maturation in human islets during human pancreas development (*Hrvatin et al., 2014*), the signals that reverse β cell de-differentiation (i.e., inhibition of Alk5 signaling) may be conserved between mouse and human.

Alk5 inhibitor II has been previously identified by Rezania et al. in an independent screen aimed at inducing functionally mature endocrine cells from human embryonic stem cells (*Rezania et al., 2011*). Ichida et al. showed that this inhibitor can replace Sox2 in cell reprogramming (*Ichida et al., 2009*). Interestingly, it was recently reported that β cells of mice carrying a conditional deletion of both Alk5 (referred to as TGFβ receptor I) and TGFβ receptor II do not proliferate in response to inflammatory cytokines (*Xiao et al., 2013*). In our results, Alk5 inhibitor II restored specific β cell gene expression in de-differentiated β cells, blocked cytokine-induced β cell stress, and stimulated over-expression of these genes in β cells from healthy, non-diabetic mice, and humans. It is noteworthy that the up regulation of SMAD7, a downstream mediator of TGFβ signaling, promotes β cell proliferation (*Xiao et al., 2014*). Inhibition of Alk5 would inhibit SMAD7-induced proliferation. This may hint on the intriguing idea that β cell proliferation, at least under inflammatory stress, may require a phase of de-differentiation. Taken together, these results suggest that Alk5 signaling may be constitutively active in β cells, that sustaining mature β cell phenotype depends on constant inhibition of this signal, and that the inhibition of Alk5 signaling may confer its effect by inducing expression of β cell transcription factors including *MafA*, *Nkx6.1*, and *Pdx1*. This postulated inhibition of Alk5 signaling in mature β cells develops during the first postnatal weeks, when the cells reach their fully mature state (*Blum et al., 2012*; *Szabat et al., 2010*; *Szabat et al., 2011*), and is reduced under diabetic stress or when the cells are taken out of their proper niche and grown in vitro. If inhibition of Alk5 signaling is not restored, the β cells will evidently de-differentiate and disappear. Alk5 is a broadly expressed protein and its activity is required in many other tissues besides β cells. Indeed, our preliminary attempts to inject high dosage of Alk5 inhibitor II to diabetic mice resulted in overall poor health without significant reduction of blood glucose. We therefore suggest that screening for compounds that inhibit Alk5 signaling specifically in β cells may yield compounds that in combination with traditional blood-glucose lowering medicines will delay, prevent, or perhaps restore the loss of healthy, mature β cell function in T2D patients.

## Materials and methods

### Animals

Animal experiments were performed in compliance with the Harvard University International Animal Care and Use Committee (IACUC) guidelines. Mouse strains used were C57BL/6, Lep[Ob/Ob] (*Zhang et al., 1994*), Lepr[Db/Db] (*Chen et al., 1996*), Ins2[Akita] (*Wang et al., 1999*), Insulin2-Cre transgenic mice (*Postic et al., 1999*), *Ucn3*-GFP transgenic mice (*Gong et al., 2003*), SCID-beige mice, R26H2BCherry mice, and RCU mice. R26H2BCherry mice (carrying a floxed nuclear-labeling reporter composed of histone H2B fused mCherry) were generated by genetic targeting of the Rosa26 locus of V6.5 mouse ES cells with the construct Rosa26-Puro-p (A)-CAGS-lox-PGK:neo-p (A)-lox-H2BCherry-p (A). Targeted ES cells were injected into BDF1xB6 blastocysts, and germline transmission was detected through breeding of chimeras with C57BL/6 females. To generate RCU mice, mice homozygous for both R26H2BCherry and *Ucn3*-GFP were crossed with homozygous Insulin2-Cre mice. All RCU progeny are triple hemizygous at all three alleles. Induction of transient insulin resistance by S961 was done with an osmotic pump as previously described (*Yi et al., 2013*). Blood glucose levels were measured in non-fasted animals using OneTouch Ultra2 glucometer (LifeScan, Milpitas, CA). For islet isolation, adult pancreata were perfused through the common bile duct with 0.8 mM Collagenase P (Roche), and fetal

and neonatal pancreata were dissected wholly without perfusion. Pancreata were digested with 0.8 mM Collagenase P (Roche) and purified by centrifugation in Histopaque gradient (Sigma).

## Immunostaining

Pancreata were fixed by immersion in 4% paraformaldehyde overnight at 4°C. Samples were washed with PBS, incubated in 30% sucrose solution overnight, and embedded with optimal cutting temperature compound (Tissue-Tek). 10-μm sections were blocked with 10% donkey serum (Jackson Immunoresearch) in PBS/0.1% Triton X and incubated with primary antibodies overnight at 4°C. Secondary antibodies were incubated for 1 hr at room temperature. Antibodies and dilutions used include rabbit anti-mouse Ucn3 (1:600-1:800, Phoenix Pharmaceuticals), Guinea Pig anti-insulin (1:800, DAKO), Alexa Fluor 488 donkey anti-rabbit (1:400, Invitrogen), and DyLight 649 donkey anti-guinea pig (1:400, Jackson Immunoresearch). Nuclei were visualized with DAPI. Images were taken using an Olympus IX51 Microscope or Zeiss LSC 700 confocal microscope.

## Automated screen

Islets from adult RCU mice were isolated and plated on 804G matrix (*Lefebvre et al., 1998*) for 1 week in a 384-well plate format. Compound libraries were added on day 7, and islets were cultured for an additional week in the presence of compounds. Each compound was tested in duplicates of two or three concentrations. A list of all compounds and concentrations appears in *Supplementary files 1*. Fresh un-manipulated RCU islets were used as a positive control, and DMSO- or untreated islets were used as a negative control. The islets were fixed on day 11 for automated image acquisition and analysis using a Cellomics ArrayScanVTI. Cell nuclei of target cells were identified by nuclear mCherry expression, and a 2 pixel cytoplasmic mask was drawn around each nucleus. The GFP fluorescence in the cytoplasmic mask of freshly isolated islets was used as a control to identify fluorescence intensity thresholds that enabled automated calls on each individual cell. Cells that displayed GFP fluorescence equal or greater than found in control cells were identified as being positive for the *Ucn3*-GFP reporter. Percentages of mCherry positive cells that co-express GFP were calculated for each well and used to identify conditions that significantly increased the number of GFP positive cells over negative controls. Positive hits are selected according to their statistical significance (p value by *t* test) over the negative control.

## Quantitative real-time PCR and cytokine treatment

Total RNA from fresh or cytokine-treated whole islets was isolated using RNeasy Plus Mini Kit (Qiagen). cDNA was prepared with random primers using SuperScript III reverse transcriptase (Life Technologies). For cytokine treatment, isolated islets were recovered overnight in islet media (DMEM containing 1gr/l glucose, 10% vol/vol FBS, 0.1% vol/vol Penicillin/Streptomycin), followed by 24-hr incubation with 10 ng/ml of either mouse IL-1β, mouse TNFα or mouse INFγ (R&D Systems), with the addition of Alk5 inhibitor II (1 μM, Axxora) or vehicle (DMSO) at the same dilution. Relative expression of *Ucn3*, *Ins1*, *Nkx6.1*, *Pdx1*, and *FoxO1* was determined using gene-specific TaqMan probes with TaqMan Fast Universal PCR Master Mix (Life Technologies) on an ABI 7900 Real-Time PCR machine. Relative expression of mouse *MafA* was determined using Brilliant III Ultra-Fast SYBR Green QPCR Master Mix (Agilent) on the same machine. Primers for mouse *MafA* were 5′-AGCGGCACATTCTGGAGAG-3′ forward and 5′-TTGTACAGGTCCCGCTCCTT-3′ reverse. Levels of gene expression were normalized to the expression of Ubc or Eif2A genes.

## Human islets

Institutional review board approval for research use of human tissue was obtained from the Harvard University Faculty of Arts and Sciences. Human islets were obtained from NDRI (The National Disease Research Interchange). Donor anonymity was preserved, and the human tissue was collected under applicable regulations and guidelines regarding consent, protection of human subjects and donor confidentiality. Human islets were grown in CMRL 1066 Supplemental medium (Mediatech), 10% vol/vol HyClone FBS (Thermo Scientific), 1% vol/vol Penicillin/Streptomycin (Corning Cellgro) for 4 days before treatment for 24 hr with Alk5 inhibitor II (Axxora).

## Acknowledgements

We are grateful to Yoav Mayshar, Ayal Ben-Zvi, and all members of the Melton laboratory and the Harvard Medical School Critical Discussion Group for helpful discussions and experimental advice. We thank Peng Yi and Ji-Sun Park for samples of diabetic mice, and to Dena Cohen and Jenny Ryu for help

with mouse transplantations. BB and QPP are supported by Juvenile Diabetes Research Foundation post-doctoral fellowships. OB is supported by Howard Hughes Medical Institute. DAM is an Investigator of the Howard Hughes Medical Institute.

## Additional information

### Funding

| Funder | Author |
| --- | --- |
| Howard Hughes Medical Institute | Ornella Barrandon, Douglas A Melton |
| Juvenile Diabetes Research Foundation International | Barak Blum, Quinn P Peterson |

The funders had no role in study design, data collection and interpretation, or the decision to submit the work for publication.

### Author contributions

BB, Conception and design, Acquisition of data, Analysis and interpretation of data, Drafting or revising the article; ANR, OB, ACA, LSD, JCD, QPP, Acquisition of data, Drafting or revising the article; RM, Acquisition of data, Contributed unpublished essential data or reagents; LLR, Conception and design, Drafting or revising the article; DAM, Conception and design, Analysis and interpretation of data, Drafting or revising the article

### Ethics

Human subjects: Institutional review board approval for research use of human tissue was obtained from the Harvard University Faculty of Arts and Sciences. Human islets were obtained from NDRI (The National Disease Research Interchange). Donor anonymity was preserved, and the human tissue was collected under applicable regulations and guidelines regarding consent, protection of human subjects and donor confidentiality.

Animal experimentation: Animal experiments were performed in compliance with the Harvard University International Animal Care and Use Committee (IACUC) guidelines (protocol #93-15).

## Additional files

### Supplementary file

• Supplementary file 1. A list of all compounds used in the automated screen. (**A**) Listed are all growth factors used in the initial screen (*Figure 4B*) (**B**) Listed are all small molecules used in the secondary screen (*Figure 4C*). Working concentration (ng/ml for growth factors and µM for small molecules), percentage of *Ucn3*-GFP positive cells and p value of all replicates are listed for each compound.

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
