## [Decision Letter]

Thank you for sending your work entitled “Reversal of β cell de-differentiation by a small molecule inhibitor of the TGFβ pathway” for consideration at *eLife.* Your article has been favorably evaluated by Fiona Watt (Senior editor), a Reviewing editor, and 2 reviewers.

The Reviewing editor and the reviewers discussed their comments before we reached this decision, and the Reviewing editor has assembled the following comments to help you prepare a revised submission.

The manuscript by Blum et al. demonstrated that de-differentiation of beta cells occurs in T2D. The authors show that Ucn3, a marker for mature beta cells, is down regulated in T2D, and thus stands as a useful marker for detecting early beta cell failure. The authors then used Ucn3 as a readout to screen for factors that reverse beta cell dedifferentiation and identified a small molecule inhibitor of TGFbeta receptor I (Alk5) that showed protective effects on beta cells and restored mature beta cell identity. They found that Alk5 inhibitor II induced expression of Maf, Nkx and Pdx in de-differentiated b cells. However, the inhibitor failed to induce insulin.

While both of the reviewers agree that this is a very interesting study, the authors need to appropriately address the reviewers' constructive comments before the manuscript is acceptable. The summaries of the reviewers' major comments are follows:

1) The authors state that Ucn3 is an early marker of de-differentiation. This is a very interesting point. On the other hand, a more detailed examination should be done to give more information on early molecular events that lead to T2D. In this manuscript, comparison of Ucn3 expression in normal vs T2D mice shows correlation of Ucn3 expression and severity of diabetes (Figure 1). However, this is not sufficient to support de-differentiation. The data showing down-regulation of Ucn3 expression precedes insulin down-regulation should be provided. In Figure 2, the authors used S961 to induce de-differentiation. However, the data shown were obtained at a single time point and it does prove Ucd3 is an early marker. In addition, no quantitative analysis was performed. Quantitative evaluation of the expression levels of Ucn3, Maf, Nkx, Pdx and insulin at different time points during de-differentiation and also re-differentiation is necessary.

2) Although de-differentiation by culture of islets reduced glucose response, the data shown are not sufficient to prove de-differentiation. At least gene expression of some of b cell genes in this culture system should be shown. To prove the effect of Alk5 inhibitor II on de-differentiated cells, the inhibitor should be added to cultured de-differentiated islets. Alternatively, the inhibitor may be administrated to T2D mice to see if it ameliorates the disease by up-regulating the b cell genes. The manuscript should be strengthened by performing in vivo functional assays to show whether the chemical, Alk5i, could restore Glucose stimulating insulin secretion (GSIS) or blood glucose, and also protein level of Ucn3 by immunostainings of Ucn3.

3) Other suggestions for revision of the Figures:

In Figure 1, the authors mixed three diabetic mice (Ob/Ob, Db/Db and Akita) and divided them together into groups of mildly and severely diabetic group and compared the expression of markers. They should show individual disease instead of mixing them. No results of Akita mice were found in Figure 1.

Figure 2–figure supplement 4: There is no unit for Y axis. From the legends, it seems that each bar represents average gene expression. The author should clarify this.

Figure 4: The GSIS of islets treated with IL1, INFga, and TNFa should be examined.

Figure 5: It is not clear to what extent the expression of the transcription factors recovered. The levels should be compared with that of the healthy BL5, or with isolated islets from healthy adult mice. It is necessary to show that Ucn3 expression is restored by treatment with Alk5i by immunohistochemistry.

4) Suggestions for revision of the text:

Authors should add some descriptions on these issues in the Discussion on the following points: As the reviewer 2 suggests, BMP9, TGFb-sRIII and Alk5 inhibitor II reversed de-differentiation. However, the mechanism of action of those factors and physiological significance of the finding remains unknown. As Alk5 inhibitor II failed to induce insulin, clinical significance of the finding is unclear. It might be worthy to comment on the recently reported observation that up-regulation of SMAD7 promotes proliferation of b cells (Xiao et at, PNAS 2014). Inhibition of Alk5 would inhibit SMAD7 up-regulation and hence it may suppress re-differentiation.

---

## [Author Response]

We thank the reviewers for their valuable remarks and comments. We have now revised the manuscript to accommodate these comments. Accordingly, we added one new figure (Figure 5), and amended the figures, figure supplements, and the main text.

*1) The authors state that Ucn3 is an early marker of de-differentiation. This is a very interesting point. On the other hand, a more detailed examination should be done to give more information on early molecular events that lead to T2D. In this manuscript, comparison of Ucn3 expression in normal vs T2D mice shows correlation of Ucn3 expression and severity of diabetes (*Figure 1*). However, this is not sufficient to support de-differentiation. The data showing down-regulation of Ucn3 expression precedes insulin down-regulation should be provided*.

Figure 1 shows immunostaining and qRT-PCR analyses of insulin and Ucn3 in three diabetic mouse models, with severity of diabetes ranging from mild to severe. The data presented demonstrate that both Ucn3 protein and Ucn3 mRNA are significantly reduced in the diabetic mice while insulin protein is still present in all models (Figure 1), and insulin mRNA levels are significantly reduced only in the most severely diabetic mice (Figure 1). This shows that the down-regulation of Ucn3 expression precedes that on insulin. We have now amended the legend to Figure 1 and the relevant text.

*In*
Figure 2*, the authors used S961 to induce de-differentiation. However, the data shown were obtained at a single time point and it does prove Ucd3 is an early marker. In addition, no quantitative analysis was performed. Quantitative evaluation of the expression levels of Ucn3, Maf, Nkx, Pdx and insulin at different time points during de-differentiation and also re-differentiation is necessary*.

We have added more data to Figure 2 showing new data from quantitative RT-PCR analysis of the reduction on Ucn3 and insulin (Figure 2) and MafA, Nkx6.1 and Pdx1 (Figure 2—figure supplement 1) at different time points during S961-induced de-differentiation and subsequent recovery following S961 withdrawal. These data confirm that Ucn3 levels are reduced to about half the normal levels as early as four days after the beginning of S961 treatment, and are down to one third by day seven. Insulin mRNA levels, on the other hand, are not significantly reduced by this time. This supports the conclusion that Ucn3 is an early marker of S961-induced de-differentiation. Nkx6.1 and Pdx1 and MafA mRNA levels are also reduced at the same time as Ucn3, to about the same magnitude, supporting the notion that the phenomenon represents β cell de-differentiation or at least a loss of the mature β cell state. These new results are discussed in the main text.

*2) Although de-differentiation by culture of islets reduced glucose response, the data shown are not sufficient to prove de-differentiation. At least gene expression of some of b cell genes in this culture system should be shown. To prove the effect of Alk5 inhibitor II on de-differentiated cells, the inhibitor should be added to cultured de-differentiated islets*.

We have now added a new figure (Figure 5), dedicated to culture-induced de-differentiation and Alk5i-induced re-differentiation. Figure 5 thus show live insulin-lineage tracing mCherry and Ucn3-GFP fluorescence in RCU islets cultured on 804G for two weeks. These data show that addition on Alk5i to the islets after one week of de-differentiation restores Ucn3-GFP fluorescence to levels comparable to that of fresh islets. Quantitative RT-PCR analysis of FACS-sorted mCherry-labelled β cells from these cultures shows that the mRNA levels of Ucn3, MafA, Nkx6.1, Pdx1 and Insulin1 are diminished in the untreated cultures, supporting the idea of β cell de-differentiation, i.e., the loss of β cell mature gene expression. Strikingly, addition of Alk5i one week after the start of de-differentiation is sufficient to restore the expression levels of Ucn3, MafA, Nkx6.1 and Pdx1, and to a lesser extent insulin1. Figure 5 shows by immunostaining the reduction of Ucn3 and insulin proteins in the de-differentiated cultures, and their recovery following Alk51 treatment.

*Alternatively, the inhibitor may be administrated to T2D mice to see if it ameliorates the disease by up-regulating the b cell genes. The manuscript should be strengthened by performing in vivo functional assays to show whether the chemical, Alk5i, could restore Glucose stimulating insulin secretion (GSIS) or blood glucose, and also protein level of Ucn3 by immunostainings of Ucn3*.

We agree that our conclusion would be strengthened by performing in vivo experiments with Alk5i, and we have indeed attempted to do this. Unfortunately, Alk5 is a broadly expressed gene with significant roles in many tissues, and our attempts to inject high doses of the chemical into diabetic mice resulted in poor general health and no apparent amelioration of blood glucose levels. We discuss this attempt in the Discussion section, and suggest that screening for mediators of Alk5 signaling that are expressed specifically in β cells, or at least are not as crucial as Alk5 itself, may yield potent drugs for manipulating this pathway in vivo.

*3) Other suggestions for revision of the Figures*:

*In*
Figure 1*, the authors mixed three diabetic mice (Ob/Ob, Db/Db and Akita) and divided them together into groups of mildly and severely diabetic group and compared the expression of markers. They should show individual disease instead of mixing them. No results of Akita mice were found in*
Figure 1.

We have added a panel in Figure 1 (panel B), showing Ucn3 and insulin1 expression for each disease model. We have also added immunostained images for the Akita model to Figure 1.

*Figure 2–figure supplement 4: There is no unit for Y axis. From the legends, it seems that each bar represents average gene expression. The author should clarify this*.

We thank the reviewers for noting this mistake in the legend (now Figure 3—figure supplement 3). The data shown in this figure represent GSIS results of fresh islets vs. islets de-differentiated on 804G. We have now corrected the figure legend and added Y axis titled “insulin (ng/ml)”.

Figure 4*: The GSIS of islets treated with IL1, INFga, and TNFa should be examined*.

We have added GSIS results for islets treated with the three cytokines. This in now shown in Figure 6—figure supplement 1.

Figure 5*: It is not clear to what extent the expression of the transcription factors recovered. The levels should be compared with that of the healthy BL5, or with isolated islets from healthy adult mice. It is necessary to show that Ucn3 expression is restored by treatment with Alk5i by immunohistochemistry*.

We have amended this figure (now Figure 7) to show gene expression in islets from diabetic mice normalized to the levels of islets from wild-type C56BL/6 mice treated with DMSO only. This enables the comparison of the magnitude of gene induction by Alk5i in these models compared to wild-type mice. Immunostaining for Ucn3 and insulin proteins in de-differentiated islets as well as their recovery by Alk5i are now shown in Figure 5.

4) Suggestions for revision of the text:

*Authors should add some descriptions on these issues in the Discussion on the following points: As the reviewer 2 suggests, BMP9, TGFb-sRIII and Alk5 inhibitor II reversed de-differentiation. However, the mechanism of action of those factors and physiological significance of the finding remains unknown. As Alk5 inhibitor II failed to induce insulin, clinical significance of the finding is unclear. It might be worthy to comment on the recently reported observation that up-regulation of SMAD7 promotes proliferation of b cells (Xiao et at, PNAS 2014). Inhibition of Alk5 would inhibit SMAD7 up-regulation and hence it may suppress re-differentiation*.

The reviewers raise an excellent point. While inhibition of Alk5 by Alk5i restores β cell maturation, as is described in our manuscript, it may simultaneously have an inhibitory effect on β cell proliferation by inhibiting SMAD7. This points to the interesting idea that in order to proliferate, β cells may need to undergo some sort of de-differentiation. We now discuss this hypothesis and we have added the above citation to our references.

The down-regulation of insulin mRNA by Alk5i in these figures is intriguing, and we don’t yet have a good explanation to it. We assume that it happens because the experiments are done in low-glucose medium. We discuss this point in the text.